environmental chemistry

fir bark, activated carbon, adsorption, methylene blue

**Author for correspondence:**
Weigang Zhao
e-mail: weigang-zhao@hotmail.com

# Synthesis of activated carbon from biowaste of fir bark for methylene blue removal

Lu Luo[1], Xi Wu[1], Zeliang Li[1], Yalan Zhou[1], Tingting Chen[1], Mizi Fan[1,2] and Weigang Zhao[1]

[1]College of Material Engineering, Fujian Agriculture and Forestry University, 63 Xiyuangong Road, Fuzhou 350002, People's Republic of China
[2]College of Engineering Design and Physical Sciences, Brunel University, Uxbridge UB8 3PH, UK

(iD) WZ, 0000-0003-1804-6552

Activated carbon (AC) was successfully prepared from low-cost forestry fir bark (FB) waste using KOH activation method. Morphology and texture properties of ACFB were studied by scanning and high-resolution transmission electron microscopies (SEM and HRTEM), respectively. The resulting fir bark-based activated carbon (ACFB) demonstrated high surface area ($1552 \, \mathrm{m^2 \, g^{-1}}$) and pore volume ($0.84 \, \mathrm{cm^3 \, g^{-1}}$), both of which reflect excellent potential adsorption properties of ACFB towards methylene blue (MB). The effect of various factors, such as pH, initial concentration, adsorbent content as well as adsorption duration, was studied individually. Adsorption isotherms of MB were fitted using all three nonlinear models (Freundlich, Langmuir and Tempkin). The best fitting of MB adsorption results was obtained using Freundlich and Temkin. Experimental results showed that kinetics of MB adsorption by our ACFB adsorbent followed pseudo-second-order model. The maximum adsorption capacity obtained was $330 \, \mathrm{mg \, g^{-1}}$, which indicated that FB is an excellent raw material for low-cost production of AC suitable for cationic dye removal.

## 1. Introduction

As urbanization and industrialization advance, the environmental problem has become increasingly prominent, especially the pollution of water resources, which seriously affects water quality [1]. Among the pollutants, synthetic dyes (i.e. methylene blue (MB)) have drawn much attention because of their wide application in dyeing, textiles, printing, leather as well as in the coating industries, which causes water contamination [2,3]. Meanwhile, coloured dye wastewater is complex in nature, most of which is toxic, mutagenic and carcinogenic to aquatic organisms, causing some health problems [4,5]. MB is a very commonly used

synthetic dye (in wood, silk, leather and cotton processing) and, as a result, is often found in industrial wastewater. It belongs to the group of cationic dyes. The ingestion of water with MB into the human body can lead to health problems such as shock, diarrhoea, jaundice, etc. [5,6].

Based on the problems, several technologies, including flotation [7], aerobic and anaerobic treatment [8], micro-and ultra-filtration [9], ion exchange [10], microbial electrochemical technologies [11], oxidation [12,13] and adsorption, have been employed for wastewater treatment [14–18]. Among these methods, adsorption has received extensive attention since it is easier, cheaper, more efficient and economical than others. Thus, different adsorbents have been developed and applied to neutralize dyes and other organics in wastewater. Nguyyen & Juang [19] prepared graphene oxide/titanate nanotube compound and when applied for adsorption of MB, the adsorption capacity was low, only $26 \, mg \, g^{-1}$. Yang *et al.* [20] synthesized the graphite oxide using a kitchen microwave oven and the adsorption capacity of MB was $170 \, mg \, g^{-1}$. Dehghani *et al.* [21] used a new composite made up of shrimp waste chitosan and zeolite as adsorbent to remove MB, and the adsorption capacity was $24.5 \, mg \, g^{-1}$. Fu *et al.* [22] synthesized polydopamine (PDA) microspheres by oxidative polymerization method and used them as an adsorbent for the removal of MB, with the adsorption capacity reaching $90.7 \, mg \, g^{-1}$. Auta & Hameed [23] reported that Chitosan–clay composite was prepared and applied to remove MB, and the adsorption capacity was $142 \, mg \, g^{-1}$. Therefore, it is necessary to develop an efficient and environmentally-friendly adsorbent.

Activated carbon (AC) is one of the best adsorbents which is widely used because of its large surface area, excellent porosity, low density as well as high adsorption capacity towards various organic compounds [1,24–26]. ACs can be obtained from various agricultural waste- and by-products, which have received significant attention as they are low-cost, renewable and environmentally friendly [25,27]. Recently, several types of ACs were obtained using bamboo [28], palm shells [29], coconut shell [30,31], rich husk [32], sawdust [33], apricot stones [34], seeds [27], etc. as raw materials. The fir tree is one of the fastest-growing trees to be planted in large numbers throughout the world. As a common forestry waste, the tree bark is a low added value product, which is often burned as a fuel or treated as waste material [33]. In the course of fir tree use, a large amount of fir bark (FB) is produced, which causes gaseous pollution during burning.

Therefore, this work is focused on the preparation of AC from FB (ACFB) using the KOH activation method. As mentioned, fir tree bark is a cost-effective and high-quantity by-product, which makes it a very promising raw material for preparing low-cost activated biocarbons. To evaluate the properties of this ACFB, we used its dye adsorption capacity towards MB as a performance criterion. When not used as a fuel or treated as waste material, the value-added applications of fir tree bark have important implications for both society and the environment. The morphology texture and pore structure of ACFB were characterized using SEM and BET analyses, respectively. MB was selected to study the adsorption capacity. The effect of contact time, adsorbent dosage, initial concentration and pH on adsorption characteristics of ACFB was studied. The fitting of adsorption isotherms and kinetics were also investigated. The results suggest the as-prepared porous carbon material from fir tree bark has great potential for MB adsorption, which is comparable or better than the samples reported in the open literature.

# 2. Material and methods

## 2.1. Reagents and materials

Chinese fir (*Cunninghamia lanceolata*) bark as raw material was sourced from a commercial plantation in Fujian province (China). KOH pellets and HCl solution (approx. 36.5%), used as received, were of analytical grade and acquired from Tianjin Fuchen Chemical Reagents Factory (Tianjin, China). We used deionized water with 18.25 MΩ cm resistance.

## 2.2. Activated carbon synthesis

For the synthesis of activated carbon (ACFB), Chinese fir was washed several times with tap water until the waste water became clear. It was dried at 103°C for 24 h, after which the bark was milled and sieved through a 10 mesh sieve to obtain powder around approximately 2 mm in size.

The synthesis procedure of AC was adapted from our earlier work as follows [35]: first, crushed FB was carbonized at 450°C (from room temperature at $5°C \, min^{-1}$ rate) under constant $N_2$ flow of $500 \, ml \, min^{-1}$. After 1 h held at 450°, the samples were slowly cooled at room temperature. The resulting product was a carbonaceous precursor, which was then mixed with KOH pellets. The mixture was placed into a nickel crucible and then into a tube furnace, which was slowly heated (at $3°C \, min^{-1}$) to 700°C under a constant

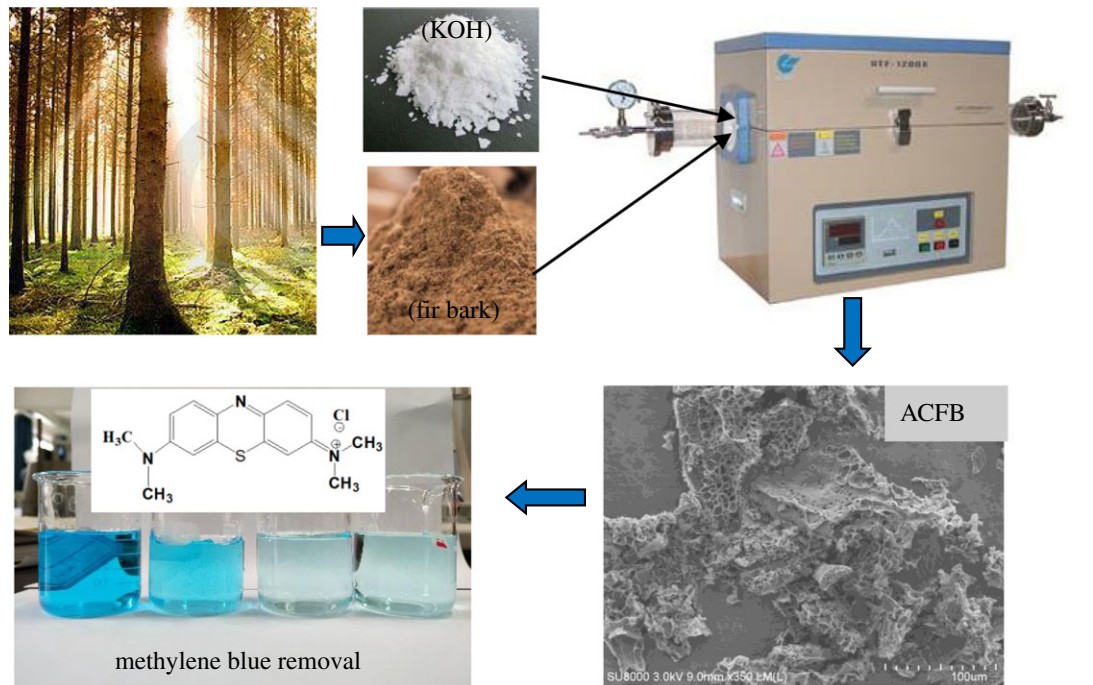

**Figure 1.** Schematic procedure of the complete synthesis route of the porous carbons from FB for MB removal.

$N_2$ flow (500 ml min$^{-1}$). After 2 h at 700°C, the samples were allowed to cool down inside the tube furnace still under nitrogen flow. The resulting AC samples were rinsed with 1 M HCl and then washed with hot water in Soxhlet for 48 h until the pH of water was stable, after which the products were dried for 24 h at 103°C. The resulting product was very pure AC. Because it was prepared from FB, it was named ACFB (figure 1).

## 2.3. Characterization methods

Specific surface area as well as pore volume and sizes of ACFB were obtained by Micromeritics ASAP 2020 automatic apparatus using nitrogen adsorption/desorption isotherms at −196°C. ACFB was degassed at 250°C under vacuum overnight. The average micropore diameters ($L_0$) and pore size distributions (PSD) were calculated using density functional theory (DFT). The surface morphology and pore texture of ACFB were characterized using scanning transmission electron microscopy (STEM; FEG SEM Hitachi S3400, Chiyoda-ku, Tokyo, Japan) as well as high-resolution transmission electronic microscopy (HRTEM; JEM-2100, JEOL, Tokyo, Japan) at 200 kV accelerating voltage.

## 2.4. Adsorption of dyes on methylene blue

Adsorption experiments were performed using the batch adsorption method to determine the influence of pH (in the 3–11 range), initial adsorbent content (in the 20–80 mg l$^{-1}$ range), ACFB dose (1–50 mg) and adsorption duration (5–180 min) on the adsorption result. For each experiment, a certain amount of ACFB was placed into a 250 ml conical flask containing 100 ml of MB of a specific concentration and at certain pH. The mixture was stirred in an orbital shaker at 30°C at 150 rpm over a specific time. Remaining MB content was measured by UV–vis UV-6300, MAPADA spectrophotometer at 664 nm maximum wavelength.

The percentage of MB adsorbed was determined based on the following formula:

$$\text{Removal } (\%) = \frac{C_0 - C_e}{C_0} \times 100. \tag{2.1}$$

The maximum MB uptake $q_e$ (in mg g$^{-1}$) was calculated as shown below

$$q_e = \frac{C_0 - C_e}{W} \times V, \tag{2.2}$$

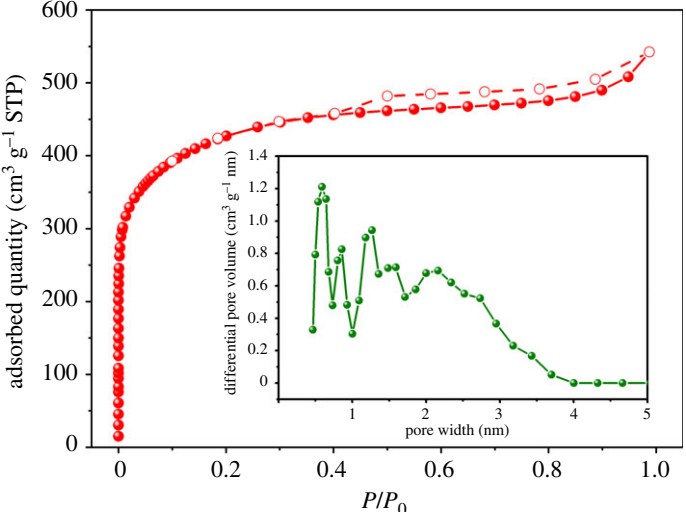

**Figure 2.** Nitrogen sorption (filled symbols)/desorption (empty symbols) isotherms and PSD (inset) for ACFB.

where $C_0$ and $C_e$ are initial and final MB concentrations in mg l$^{-1}$, respectively; $W$ is the amount of ACFB (in g) and $V$ is the volume of MB solution (in l).

# 3. Results and discussion

## 3.1. Characterization of carbon material

$N_2$ nitrogen adsorption/desorption isotherms at $-196°C$ as well as PSDs of ACFB are shown in figure 2. The nitrogen adsorption amount increased obviously at $P/P_0$ below 0.05, which indicated the mainly microporosity of ACFB [35,36]. ACFB isotherms in the 0.4–0.99 $P/P_0$ range belong to type I isotherm with H4 hysteresis according to the IUPAC classification. Such isotherms are typical for materials with a wide pore distribution, including mesopores, which were present in our ACFB [35,37]. PSD obtained using DFT calculations also showed a wide range: from 0.5 to 4 nm (figure 2 (inset)), which confirmed the results above. BET analysis showed that the surface area of ACFB was as high as 1552 m$^2$ g$^{-1}$, and its micro- and mesopore volumes were 0.56 and 0.28, respectively. It was found that the proportion of microporosity to total porosity, $V_{DR}/V_{0.99}$, is 0.68, demonstrating that ACFB is mainly microporous with a small portion of mesopores.

The SEM and TEM images of the ACFB sample in figure 3a,b reveal a great number of the pores (micrometre and nanometre in size), which were formed during the carbonization and activation processes. Pore system and excellent pore morphology observed by SEM and TEM agree with those obtained from BET analysis. Such porosity should definitely provide ACFB with high adsorption capacity towards MB.

The TGA, FTIR, XPS spectra and elemental analysis were carried out to investigate the thermostability and surface properties of ACFB in our previous work [24,35]. Upon heating to high temperature, pyrolysis of organic substances produces volatile products, which means that most of the non-carbon elements, hydrogen, nitrogen and oxygen are removed in gaseous form by pyrolytic decomposition, and leave a solid residue enriched in carbon [35]. The results of XPS spectra and elemental analysis also respond to the conclusion that the obtained AC is pure [24].

## 3.2. Effect of adsorption process parameters on the removal of MB

### 3.2.1. Effect of adsorption duration and contact time between MB and ACFB

The contact time is a non-negligible parameter for the MB adsorption process. An appropriate contact time cannot only improve the treatment efficiency but also provide the most cost-effective route. Thus, the effect of contact time of MB adsorption on to ACFB was tested at 30°C with 10 mg of ACFB, 100 ml of 20 mg l$^{-1}$ as initial MB concentration and at pH = 7. Adsorption was allowed to proceed for 180 min to determine the optimum adsorption time (see figure 4 for the results). The results revealed that MB adsorption rate increased rapidly. MB removal continued almost linearly during the initial

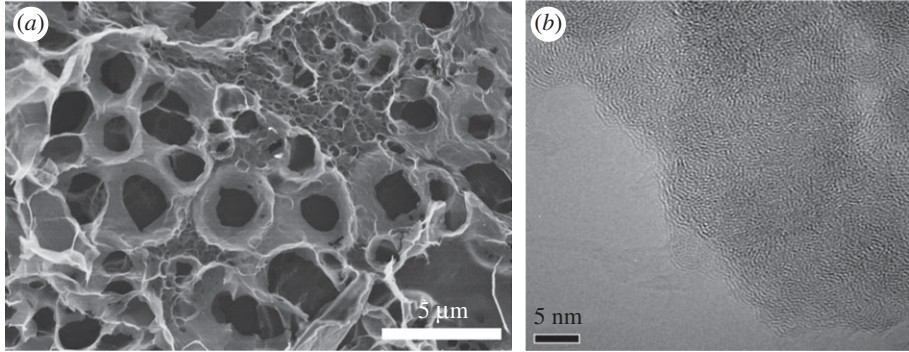

**Figure 3.** SEM (*a*) and TEM (*b*) images of ACFB.

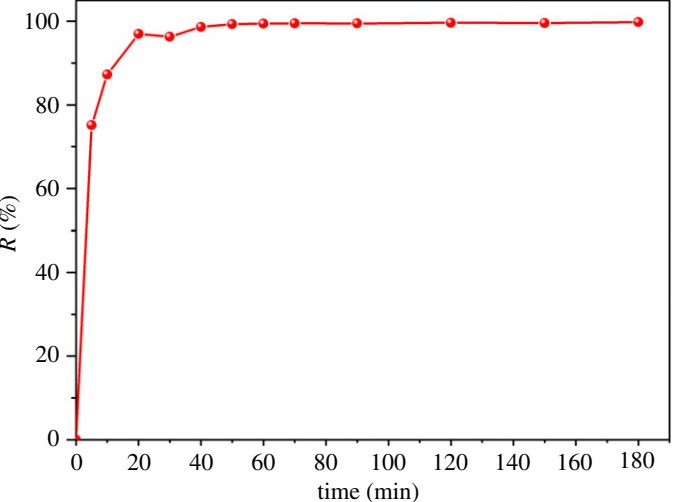

**Figure 4.** Effect of the contact time on the removal of MB ($C_0$ = 20 mg l$^{-1}$, *m* = 10 mg, *T* = 30°C, pH = 7, *v* = 100 ml).

contact period, and then gradually slowed down until equilibrium was established at around 20–40 min. Such behaviour was observed because of the higher availability of more active sites on the ACFB surface as well as the weak internal diffusion resistance during early adsorption stages. After the early stages, a plateau formed, during which only an insignificant increase could be seen, mostly because MB content in the solution was significant as active sites were already saturated and diffusion into the ACFB surface pores slowed down [34,38,39].

### 3.2.2. Effect of activated carbon dosage

The initial amount of an adsorbent is of high significance for adsorption processes. The dosage of the adsorbent at the beginning of the adsorption process affects the total amount of available pores, which will affect the overall adsorption rate and total MB amount adsorbed by ACFB [1,5]. Thus, as initial amounts we used 1, 2.5, 5, 10, 20 and 50 mg of ACFB. All other adsorption parameters were the same: 100 ml of 20 mg l$^{-1}$ MB solution, 30°C, pH = 7 and 180 min equilibration time. Results showing MB removal rates as a function of the initial ACFB content are shown in figure 5. The percentage of removed MB increased dramatically as the weight of the initial ACFB increased: MB removal efficiencies increased from 3.11% at a dosage of 1 mg to 99.78% at a dosage of 10 mg. This can be attributed to the large available surface area as well as abundant active sites for MB molecules to adsorb [40,41].

### 3.2.3. Effect of the initial concentration of MB on its adsorption

Eight different concentrations (20, 25, 30, 35, 40, 50, 60 and 80 mg l$^{-1}$) of MB were chosen to study how initial dye MB content affected its adsorption on ACFB. All other experimental adsorption parameters

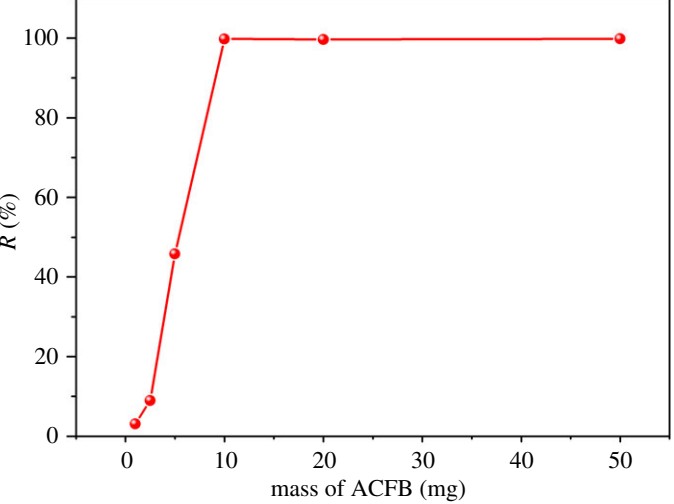

**Figure 5.** Effect of the AC doses on the removal of MB ($C_0 = 20$ mg $l^{-1}$, $t = 180$ min, $T = 30°C$, pH $= 7$, $v = 100$ ml).

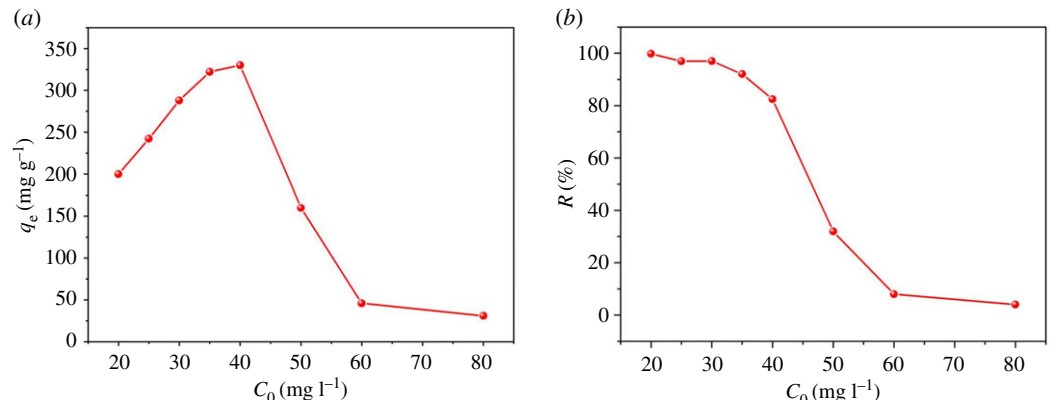

**Figure 6.** Effect of initial dye concentration of MB on maximum dye uptake (*a*) and the percentage removal by ACFB (*b*) ($m = 10$ mg, $t = 180$ min, $T = 30°C$, pH $= 7$, $v = 100$ ml).

were the same: including 100 ml of MB, pH $= 7$, 10 mg of ACFB, 30°C and 150 rpm agitation speed. As can be seen from figure 6*a*, adsorption curves display two stages of dye uptake as initial MB concentration increases from 20 to 40 mg $l^{-1}$: the first one demonstrates constantly increasing, and the second one shows decreasing adsorption capacities [42,43]. At a relatively lower dye concentration, higher dye content concentration will increase the effective contact area between dye molecules and ACFB. It will also provide the necessary driving force to overcome MB mass transfer resistance on the interface, which drives adsorption to higher capacity values [16,40]. At MB concentrations above 40 mg $l^{-1}$, a majority of the active sites are consumed. Therefore, MB adsorption slows down, and a lot of MB in the second phase remains in the solution, which was seen during the second of our adsorption tests [34,44]. Figure 6*b* displays decreased removal constant as initial MB concentration increases. Lower consumption of MB at its higher concentrations was because of high MB/active sites ratio. The ACFB surface quickly becomes saturated with MB at high concentrations, implying the dependence of adsorption on MB initial concentration [42].

### 3.2.4. pH effect

pH is also considered as one of most essential factors affecting adsorption processes, mostly because it affects adsorbent surface charge. To test how pH affects MB adsorption processes on ACFB, we performed adsorption experiments in the wide pH range and at 20 mg $l^{-1}$ initial MB concentration, 10 mg ACFB dose, at 30°C and 60 min equilibration time. MB removal percentage increased slightly as pH values increased from 3 to 11 (figure 7). At low pH, abundant $H^+$ compete with MB cations for the active sites. Thus, MB adsorption on ACFB becomes inhibited at low pH vales. Therefore,

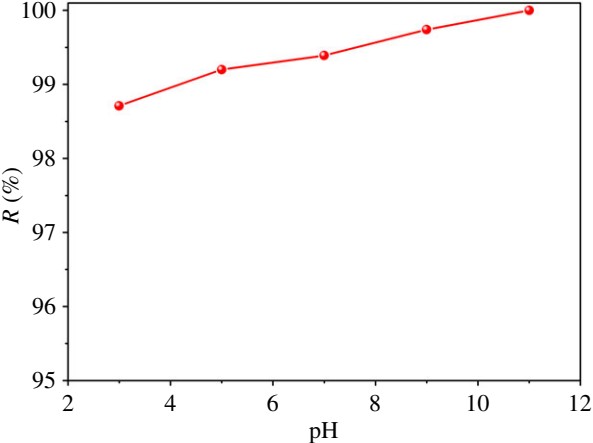

**Figure 7.** Effect of the pH on the removal of MB ($C_0 = 20$ mg $l^{-1}$, $m = 10$ mg, $t = 60$ min, $T = 30°C$, $v = 100$ ml).

as mid-level pH values, there are less competitive protons in the solution, which is beneficial for MB adsorption on the ACFB surface [45]. However, our results showed over 98% removal percentage values of MB in both acidic and neutral pH values. This is due to the fact that at low pH, MB remains at cationic and molecular form and can enter into the pores of the adsorbent surface very easily. With increasing pH, the surface of the adsorbent becomes more negative, making it favourable for the cationic dye adsorption [40]. Thus, MB adsorption on ACFB is governed not only by electrostatic interactions but also by van der Waals attraction, $\pi$- and other chemical interactions between MB and ACFB surface [38]. In order to get a deep insight in the surface chemical properties of ACFB, the FTIR spectra and XPS analysis were conducted in the previous work [35]. The existence of functional groups such as –COOH, –OH and –NH, on the surface of ACFB, suggests that the carbon material is CxOH, where Cx = carbon. It is necessary to note that the hydroxylated surface groups vary at different pH values because of the protonation/deprotonation processes (i.e. $CxOH + H^+ \leftrightarrow CxOH_2^+$ at low pH, and $CxOH \leftrightarrow CxO^- + H^+$ at high pH) [45].

## 3.3. Adsorption isotherms

Adsorption isotherms obtained in this work were fitted using the Langmuir, Freundlich and Tempkin models. Their correlation with our adsorption processes was judged by the values of correlation coefficient ($R^2$) and errors.

The Langmuir model assumes monolayer adsorption on a homogeneous surface with all active sites being equivalent and with the same energy. The Langmuir model also assumes dynamic equilibrium and no interaction between adsorbates [43]. It is typically described by the following formula [46]:

$$\frac{C_e}{q_e} = \frac{C_e}{q_{max}} + \frac{1}{q_{max}K_L},$$ (3.1)

where $q_e$ is the amount of adsorbed dye at the equilibrium (in mg $g^{-1}$), $q_{max}$ correlates with the maximum monolayer adsorption capacity (in mg $g^{-1}$), $K_L$ is an adsorption constant describing affinity between MB and ACFB (in l $mg^{-1}$) and $C_e$ is the MB equilibrium concentration.

The Freundlich model is described by a formula assuming heterogeneous multilayer adsorption on heterogeneous surfaces. The Freundlich model also assumes interactions between the adsorbates and that adsorption capacity increases with the analyte concentration. The formula describing the Freundlich model is shown below [47]

$$\log q_e = \frac{1}{n_F}\log C_e + \log K_F,$$ (3.2)

where $K_F$ is the reaction constant reflecting adsorption capacity (in l $mg^{-1}$), and $1/n_F$ indicates the dimensionless exponent of the Freundlich model to show adsorption intensity (it is calculated from the slope and an intercept of $\log q_e$ versus $\log C_e$ plot).

The Temkin adsorption assumes a decrease in adsorption heat because of the adsorbent/adsorbate interaction as coverage with molecular layers increases. Mathematically, it can be expressed as [40,48]

$$q_e = B\ln K_T + B\ln C_e,$$ (3.3)

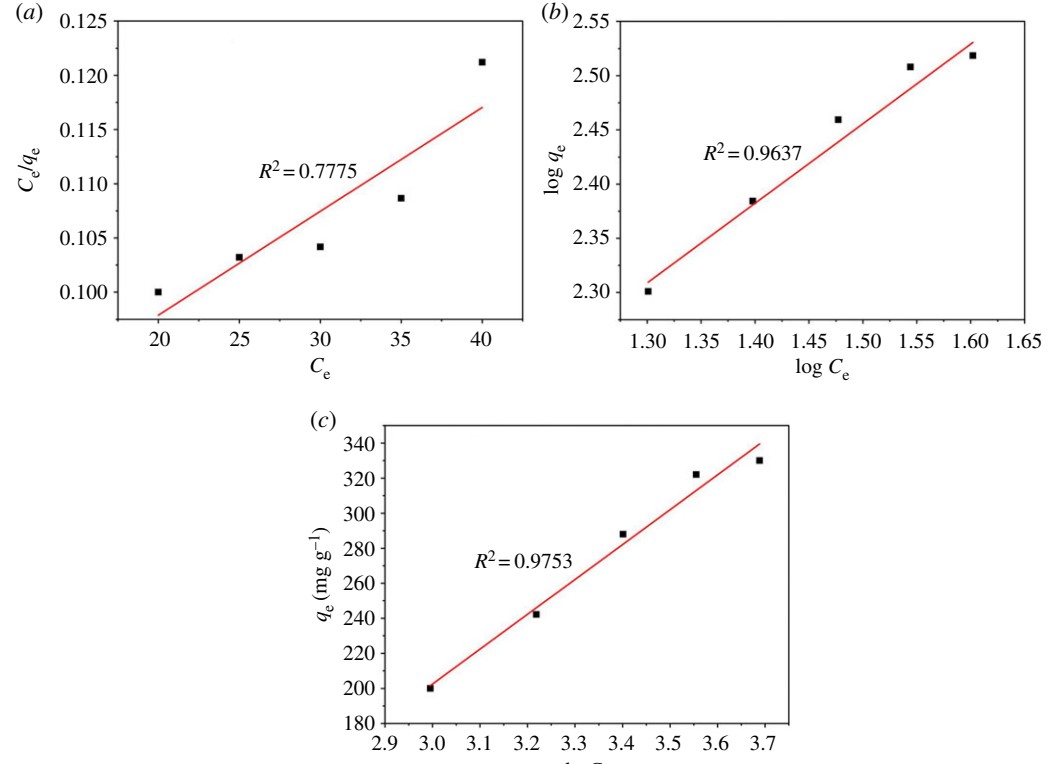

**Figure 8.** Plots of Langmuir (*a*), Freundlich (*b*) and Temkin (*c*) isotherm models for the adsorption of MB into ACFB.

**Table 1.** Adsorption isotherm parameters for MB on ACFB.

| model | | | | | | | | |
|---|---|---|---|---|---|---|---|---|
| Langmuir constants | | | Freundlich constants | | | Temkin constants | | |
| $q_{max}$ | $K_L$ | $R^2$ | $1/n$ | $K_F$ | $R^2$ | $B$ | $K_T$ | $R^2$ |
| 1044.404 | 0.01216 | 0.778 | 0.758 | 21.086 | 0.964 | 199.006 | 0.138 | 0.975 |

where $B = RT/b$, $b$ is the Temkin constant related to the adsorption heat (in $J\,mol^{-1}$), $K_T$ ($l\,mg^{-1}$) is the equilibrium adsorption constant, $R$ is the gas constant equal to $8.314\,J\,K\,mol^{-1}$ and $T$ (K) is the absolute temperature.

Figure 8 displays the Langmuir, Temkin and Freundlich isotherms for our adsorption experiments. The calculated parameters for all these isotherms along with $R^2$ values are shown in table 1. The correlation coefficient ($R^2$) for the linear portion of the Temkin model is the closest to 1.0. Thus, the Temkin model describes MB adsorption on ACFB the best. Meanwhile, it reveals this adsorption is not a monolayer adsorption process. The correlation coefficient $R^2$ for the Freundlich model was above $R^2$ obtained by fitting the Langmuir model to our data, which very likely indicates that MB adsorption on ACFB does not occur in a monolayer fashion on a homogeneous surface but rather on a heterogeneous one [3]. The value of $1/n$ equal to 0.758 is less than 1, which indicates favourable adsorption conditions [5,6,49].

## 3.4. Kinetics studies

Adsorption kinetics studies the relationship between adsorption capacity and reaction time. Thus, its main concern is adsorption speed, dynamic equilibrium, mass transfer and diffusion rates. Analysis of these parameters helps to understand adsorption process rates as well as adsorption mechanism.

For adsorption kinetics study, in this work, we used 0.01 g of adsorbent and 100 ml of 20 mg l$^{-1}$ MB solution, and then placed them into a 250 ml beaker at 30°C. We examined the adsorption rate and MB

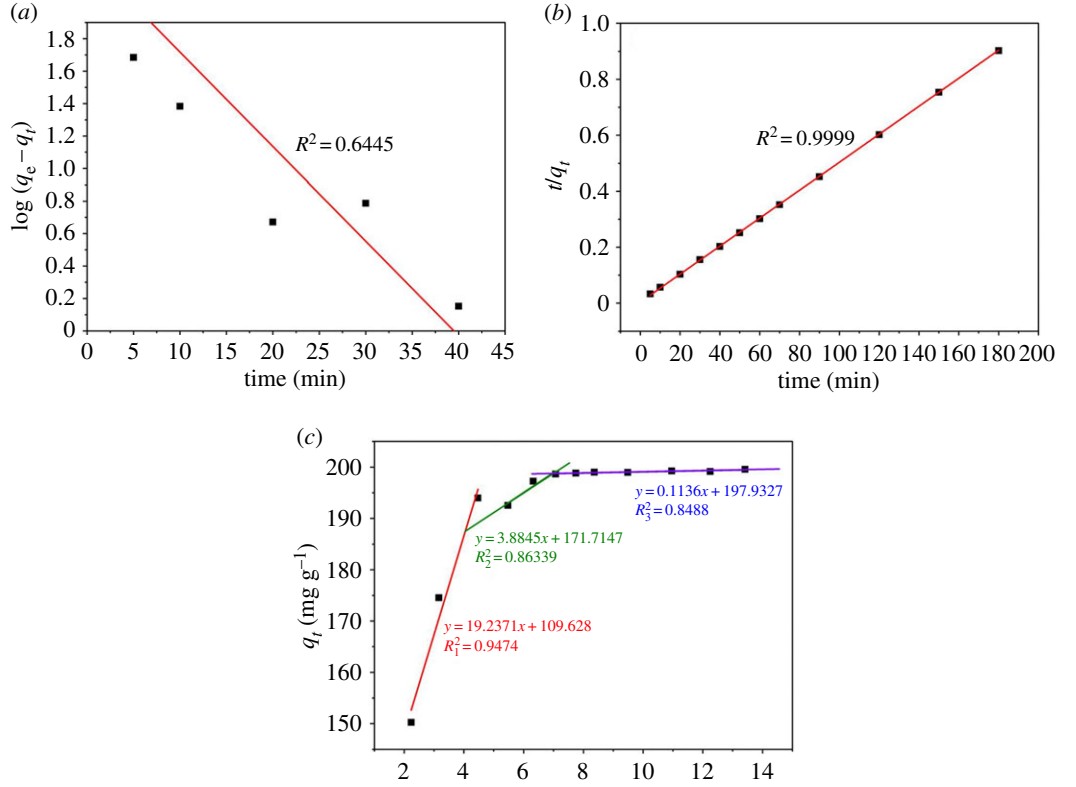

**Figure 9.** Plots of pseudo-first order (*a*), pseudo-second order (*b*) and intraparticle diffusion (*c*) for the adsorption of MB into ACFB.

removal mechanism by ACFB using different equilibrium times (5–180 min) to understand and develop a solid/liquid-phase equilibrium kinetic model.

Our experimental data were fit using the pseudo-first and -second-order reaction models as well as an intraparticle diffusion model with the goal of establishing adsorption rates.

The pseudo-first-order model is mathematically described as shown below [50]

$$\log(q_e - q_t) = \log q_e - K_1 t, \tag{3.4}$$

where $k_1$ is the pseudo-first-order kinetic constant (in $1/\text{min}^{-1}$) and $t$ is the time (in min).

The pseudo-second-order kinetic model can be expressed by the following equation [50,51]:

$$\frac{t}{q_t} = \frac{1}{K_2 q_e^2} + \frac{t}{q_e}, \tag{3.5}$$

where $k_2$ is the pseudo-second-order kinetic constant (in g (mg min)$^{-1}$), $q_t$ correlates with adsorption capacity at time $t$ in minutes (in mg g$^{-1}$).

Our experimental data were also treated using the intraparticle diffusion model to understand the diffusion process of MB on ACFB particles. It is defined as [50]

$$q_t = K_p t^{1/2} + C, \tag{3.6}$$

where $K_p$ is the intraparticle diffusion constant (in mg g$^{-1}$ min$^{0.5}$) and $C$ is the thickness of the boundary layer. At $C = 0$, intraparticle diffusion is the only controlling step. Thus, adsorption occurs inside the adsorbent. The larger the $C$, the greater the boundary layer effect on the adsorption, or in other words, the greater the effect of membrane diffusion on the adsorption process.

Figure 9 shows the three models' fitting results. Meanwhile, the detailed parameters calculated from the three kinetic models along with $R^2$ values are shown in table 2. The correlation coefficient ($R^2$) of the pseudo-second-order model was higher than the correlation coefficients obtained from other models. Compared to $q_e$ values obtained by fitting our experimental data using the pseudo-first-order and intraparticle diffusion models, the calculated $q_e$ values from the pseudo-second order show better agreement with the experimental values. Thus, taking into account all experimental data mentioned

**Table 2.** Kinetic model parameters for MB on ACFB.

| kinetic models and parameters | | MB |
|---|---|---|
| pseudo-first-order kinetics | $q_e$ | 198.67 |
| | $K_1$ | 0.058 |
| | $R^2$ | 0.6445 |
| pseudo-second-order kinetics | $q_e$ | 199.55 |
| | $K_2$ | 0.0071 |
| | $R^2$ | 0.9999 |
| intraparticle diffusion | $q_e$ | 199.55 |
| | $K_p$ | 2.99 |
| | $C$ | 169.19 |
| | $R_1^2$ | 0.4478 |
| linear fitting intraparticle diffusion | $K_{p1}$ | 19.24 |
| | $C_1$ | 109.63 |
| | $R_1^2$ | 0.9474 |
| | $K_{p2}$ | 3.88 |
| | $C_2$ | 171.71 |
| | $R_2^2$ | 0.8634 |
| | $K_{p3}$ | 0.11 |
| | $C_3$ | 197.93 |
| | $R_3^2$ | 0.8488 |

above, we determined that the pseudo-second-order kinetic model agrees the best with MB adsorption on ACFB than other kinetic models.

Figure 9$c$ displays our experimental results fitted using the intraparticle diffusion model. The $C$-value was not zero. All corresponding curves are multilinear, and three main adsorption stages can be clearly distinguished. The initial stage with slope $K_{p1}$ (19.24) shows that the dye molecules are adsorbed from the liquid phase to the external adsorbent surfaces. The second stage had $K_{p2}$ equal to 3.88, which reveals that dye molecules enter ACFB internal pores from its surface. Such a phenomenon is called intraparticle diffusion. The third stage with the $K_{p3}$ equal to 0.11 represents MB adsorption on the ACFB sites. During these stages, as MB concentration in the solution decreased, mass transfer resistance of the adsorbate increased. As a result, the diffusion process gradually slowed down, and the slope became less steep [2,14,40,52].

Table 3 shows the adsorption capacities of ACs prepared from other agricultural by-products, polymers and carbon composite materials. It is clear from these data that AC is one of the lowest cost and one of the most effective adsorbents for MB and other organics removal. What is more, as mentioned, FB is a sort of common forestry waste, which means that the cost of FB is lower than the normal biomass, polymers or GO for AC production, such as bamboo, wood, rice husk, coconut shell, graphene and so on. All these merits, along with the zero-cost and wide availability of FB, make this type of sorbent highly promising in dye adsorption.

## 3.5. Regeneration of adsorbent

According to the above analysis, MB can be easily adsorbed into ACFB under moderate conditions. In order to get further into the practical application, the regeneration property was investigated using ethanol as the eluent. As shown in figure 10, the removal capacity of MB retained 69.41% after four cycles, indicating the ACFB material possesses good regenerability and reusability when using ethanol solution [61–63].

## 3.6. Adsorption mechanism

The MB can be better removed by AC under both acid and basic conditions, indicating that the electrostatic interaction played an important role in the adsorption process [63]. At lower pH, the

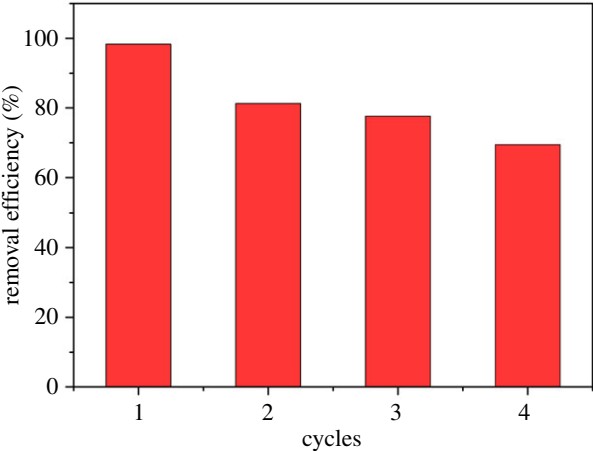

**Figure 10.** Regeneration of ACFB and MB desorption.

**Table 3.** Comparison of adsorption capacity of various adsorbents for MB.

| adsorbent | $q_e$ (mg g$^{-1}$) | ref. |
|---|---|---|
| fir bark activated carbon | 330 | this work |
| banana trunk activated carbon | 166 | [53] |
| *Acacia mangium* wood activated carbon | 158 | [51] |
| cork powder waste activated carbon | 350 | [52] |
| date pits activated carbon | 259 | [6] |
| Filtrasorb 400, | 255 | [54] |
| Norit and | 222 | |
| Picacarb granular activated carbon | 160 | |
| cotton stalk | 147 | [55] |
| CuS nanoparticle loaded on activated carbon | 208 | [56] |
| *Ephedra strobilacea* saw dust char | 31 | [57] |
| GO | 302 | [58] |
| graphene oxide/cellulose | 375 | [59] |
| graphene oxide/titanate nanotube | 26 | [19] |
| chitosan | 385 | [60] |
| graphite oxide | 170 | [20] |

various functional groups and reactive atom of dyes and adsorbent protonated and both get positive charge [40]. At higher pH, the carboxylic groups are deprotonated, and negatively charged carboxylate ligands (COO−) bind to the positively charged MB molecules. This finding confirms that the sorption of MB by FBAC is an ion exchange mechanism between the negatively and the positively charged groups [41,64]. Therefore, due to the strong repulsive force between dye and adsorbent the removal percentage decreased. The results of equilibrium and kinetic studies showed that the adsorption of MB onto FBAC was predominantly a chemisorption process [45]. Therefore, electrostatic interaction along with chemical binding between adsorbate and adsorbent mainly controlled the MB/FBAC adsorption process.

## 4. Conclusion

ACFB was prepared. It demonstrated substantial adsorption relative to MB because of its very high surface area (approx. 1552 m$^2$ g$^{-1}$) and large pore volume (approx. 0.84 cm$^3$ g$^{-1}$). The maximum MB

adsorption capacity of ACFB was 330 mg g$^{-1}$. In general, the adsorption capacity of ACFB relative to MB increased with longer equilibrium times, higher adsorbent dosage and higher initial MB concentrations. Adsorption data can be fit with a good correlation coefficient using the Freundlich and Temkin models. Adsorption kinetics followed the pseudo-second-order model. Overall, ACFB demonstrated outstanding adsorption properties relative to MB and cationic dyes in general. Thus, ACFB is very promising for use in wastewater treatment to mitigate dye pollution.

Data accessibility. The data that support the findings of this study are openly available from the Dryad Digital Repository: https://doi.org/10.5061/dryad.8tn1qg5 [65].

Authors' contributions. W.Z. and M.F. designed this work; L.L., X.W., T.C. and Z.L. performed the experiments; W.Z., L.L., Y.Z. and Z L. analysed the data; W.Z. and L.L. wrote this paper. All authors gave final approval for publication.

Competing interests. We have no competing interests.

Funding. Financial support came from the Natural Science Foundation of Fujian Province Department of Science and Technology, grant no. 2019J01386; Forestry science and technology research project of Fujian Forestry Department of China, grant no. KLB18H02A; National Natural Science Foundation of China, grant no. 31971593; Fujian Agriculture and Forestry University Fund for Distinguished Young Scholars, grant no. xjq201420.

Acknowledgements. We gratefully acknowledge the anonymous reviewers for their constructive comments which have improved the quality of this paper.

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
