## [Reviewer comments · Royal Society Open Science]

Review History

RSOS-190523.R0 (Original submission)

Review form: Reviewer 1

Is the manuscript scientifically sound in its present form?

Yes

Are the interpretations and conclusions justified by the results?

Yes

Is the language acceptable?

No

Do you have any ethical concerns with this paper?

No

Have you any concerns about statistical analyses in this paper?

No

Recommendation?

Major revision is needed (please make suggestions in comments)

Comments to the Author(s)

The manuscript "Synthesis of Activated Carbon from Biowaste of Fir Bark for Methylene Blue Removal" (RSOS-190523) prepared activated carbon from forestry fir bark. The contents fall within the scope of the journal, however, there are some issues need to be addressed properly. this manuscript need revision before publication.

Some comments are as below:

- (1) The adsorption mechanism should be stated clearly, and this part should be added in the manuscript.
- (2) The manuscript should add some comparison between AC and other materials, such as graphene, etc.
- (3) What are the advantages of AC? And, what are the highlights of this manuscript?
- (4) References Bioresource Technology, 2019, 281: 195-201 and Bioresource Technology, 2019, 276: 236-243 should be cited in this manuscript.
- (5) Some language and format issues should be revised carefully.

Review form: Reviewer 2

Is the manuscript scientifically sound in its present form?

Yes

Are the interpretations and conclusions justified by the results?

Yes

Is the language acceptable?

No

Do you have any ethical concerns with this paper?

No

Have you any concerns about statistical analyses in this paper?

No

Recommendation?

Major revision is needed (please make suggestions in comments)

Comments to the Author(s)

1. Heading Summary should be changed to Abstract.
2. For the introduction, a critical comparison of the adsorbents that were used previously for the removal of MB should be discussed to highlight their drawbacks/disadvantages that of new AC needs to be synthesized.
3. All chemicals used should be listed in sub 3.1 including the brand, city and country of origin.
4. Methylene blue should be in small letter unless used at the beginning of a sentence.
5. Sub 3.1 line 31-35 should be included in sub 3.2 as it was part of the AC synthesis procedures.
6. Sub 3.2 line 50: how can you confirm the ACFB synthesized was pure?
7. Sub 4.1: the authors should expand the characterizations to TGA and elemental analysis to prove the purity of ACFB.

8. For each parameter optimized, the author should indicate which optimum condition was chosen for the subsequent analysis. Another way is to indicate the batch adsorption method conditions in figure captions.
9. Sub 4.2.2 line 13: how is it possible to attain 100% removal?
10. For Fig 5, X-axis should be label as mass or amounts of ACFB (mg).
11. Sub 4.2.4: at the mentioned pH, what is the properties of ACFB? In this case, the change in ACFB surface needs to be discussed and correlate with ACFB properties.
12. R2 stands for correlation coefficient or non-linear regression coefficient?
13. The application of ACFB batch adsorption method for the removal of MB should be applied to the real sample to study the matrix effect and efficiency of ACFB in term of recovery.

Decision letter (RSOS-190523.R0)

25-Jun-2019

Dear Dr Zhao:

Title: Synthesis of Activated Carbon from Biowaste of Fir Bark for Methylene Blue Removal
Manuscript ID: RSOS-190523

The editor assigned to your manuscript has now received comments from reviewers. We would like you to revise your paper in accordance with the referee and Subject Editor suggestions which can be found below (not including confidential reports to the Editor). Please note this decision does not guarantee eventual acceptance.

Please submit your revised paper before 18-Jul-2019. Please note that the revision deadline will expire at 00.00am on this date. If we do not hear from you within this time then it will be assumed that the paper has been withdrawn. In exceptional circumstances, extensions may be possible if agreed with the Editorial Office in advance. We do not allow multiple rounds of revision so we urge you to make every effort to fully address all of the comments at this stage. If deemed necessary by the Editors, your manuscript will be sent back to one or more of the original reviewers for assessment. If the original reviewers are not available we may invite new reviewers.

Once again, thank you for submitting your manuscript to Royal Society Open Science and I look

forward to receiving your revision. If you have any questions at all, please do not hesitate to get in touch.

RSC Associate Editor:
Comments to the Author:
(There are no comments.)

RSC Subject Editor:
Comments to the Author:
(There are no comments.)

Reviewers' Comments to Author:
Reviewer: 1

Comments to the Author(s)

The manuscript "Synthesis of Activated Carbon from Biowaste of Fir Bark for Methylene Blue Removal" (RSOS-190523) prepared activated carbon from forestry fir bark. The contents fall within the scope of the journal, however, there are some issues need to be addressed properly. this manuscript need revision before publication.

Some comments are as below:

- (1) The adsorption mechanism should be stated clearly, and this part should be added in the manuscript.
- (2) The manuscript should add some comparison between AC and other materials, such as graphene, etc.
- (3) What are the advantages of AC? And, what are the highlights of this manuscript?
- (4) References *Bioresource Technology*, 2019, 281: 195-201 and *Bioresource Technology*, 2019, 276: 236-243 should be cited in this manuscript.
- (5) Some language and format issues should be revised carefully.

Reviewer: 2

Comments to the Author(s)

1. Heading Summary should be changed to Abstract.

2. For the introduction, a critical comparison of the adsorbents that were used previously for the removal of MB should be discussed to highlight their drawbacks/disadvantages that of new AC needs to be synthesized.
3. All chemicals used should be listed in sub 3.1 including the brand, city and country of origin.
4. Methylene blue should be in small letter unless used at the beginning of a sentence.
5. Sub 3.1 line 31-35 should be included in sub 3.2 as it was part of the AC synthesis procedures.
6. Sub 3.2 line 50: how can you confirm the ACFB synthesized was pure?
7. Sub 4.1: the authors should expand the characterizations to TGA and elemental analysis to prove the purity of ACFB.
8. For each parameter optimized, the author should indicate which optimum condition was chosen for the subsequent analysis. Another way is to indicate the batch adsorption method conditions in figure captions.
9. Sub 4.2.2 line 13: how is it possible to attain 100% removal?
10. For Fig 5, X-axis should be label as mass or amounts of ACFB (mg).
11. Sub 4.2.4: at the mentioned pH, what is the properties of ACFB? In this case, the change in ACFB surface needs to be discussed and correlate with ACFB properties.
12. R2 stands for correlation coefficient or non-linear regression coefficient?
13. The application of ACFB batch adsorption method for the removal of MB should be applied to the real sample to study the matrix effect and efficiency of ACFB in term of recovery.

Author's Response to Decision Letter for (RSOS-190523.R0)

See Appendix A.

Decision letter (RSOS-190523.R1)

05-Aug-2019

Dear Dr Zhao:

Title: Synthesis of Activated Carbon from Biowaste of Fir Bark for Methylene Blue Removal
Manuscript ID: RSOS-190523.R1

It is a pleasure to accept your manuscript in its current form for publication in Royal Society Open Science. The chemistry content of Royal Society Open Science is published in collaboration with the Royal Society of Chemistry.

Yours sincerely,
Dr Ellis Wilde
Publishing Editor, Journals

RSC Associate Editor
Comments to the Author:
(There are no comments.)

Reviewer(s)' Comments to Author:

Appendix A

Fuzhou, 06 of July 2019

Dear Editor,

We thank both you and the reviewers for the positive comments and constructive criticism concerning our manuscript. Please find here with the revised version of the manuscript '**Synthesis of Activated Carbon from Biowaste of Fir Bark for Methylene Blue Removal**' by Luo et al., which has been modified according to reviewers' feedbacks. We are still open to any criticism from you and from the reviews, and we look forward to hearing from you.

Thanks very much for your attention to our paper.

Modifications are on yellow background.

Yours sincerely,

Weigang ZHAO

Reviewer 1:

The manuscript “Synthesis of Activated Carbon from Biowaste of Fir Bark for Methylene Blue Removal” (RSOS-190523) prepared activated carbon from forestry fir bark. The contents fall within the scope of the journal, however, there are some issues need to be addressed properly. This manuscript need revision before publication.

Some comments are as below:

- (1) The adsorption mechanism should be stated clearly, and this part should be added in the manuscript.
- (2) The manuscript should add some comparison between AC and other materials, such as graphene, etc.
- (3) What are the advantages of AC? And, what are the highlights of this manuscript?
- (4) References Bioresource Technology, 2019, 281: 195-201 and Bioresource Technology, 2019, 276: 236-243 should be cited in this manuscript.
- (5) Some language and format issues should be revised carefully.

Reply to Reviewer 1:

We thank and totally agree the reviewer for the comments and constructive suggestions concerning our manuscript. We have improved our manuscript taking into account the remarks. Below are the replies for each specific comment:

- 1、 The adsorption mechanism should be stated clearly, and this part should be added in the manuscript.

We thank reviewer very much and agree with the reviewer. We have tried our best to revise and add a new part of ‘3.6 Adsorption mechanism’ which we hope meet with your approval. The modifications are on yellow background.

- 2、 The manuscript should add some comparison between AC and other materials, such as graphene, etc.

We thank and agree with the reviewer. We have done the comparison not only in the introduction part, but also in Table 3, include the various adsorbents of activated carbon, GO, polymers, some composites materials. The modifications are on yellow background.

- 3、 What are the advantages of AC? And, what are the highlights of this manuscript?

We thank reviewer very much for this valuable suggestion. We have tried our best to modify the introduction part and sub 3.4 in order to illuminate the advantages of ACs and also the highlights of this paper. The modifications are on yellow background.

4、 References Bioresource Technology, 2019, 281: 195-201 and Bioresource Technology, 2019, 276: 236-243 should be cited in this manuscript.

We thank and agree with the reviewer that there are many related articles published. We have added the references as follows in the manuscript:

- Chen J, Wang Y, Liu Y, Tang M, Wang R, Tian Y, Jia C. 2019 Bacterial community shift and antibiotics resistant genes analysis in response to biodegradation of oxytetracycline in dual graphene modified bioelectrode microbial fuel cell. *Bioresource Technol.* 276, 236-243. (DOI: 10.1016/j.biortech.2019.01.006)
- Chen J, Yang Y, Liu Y, Tang M, Wang R, Zhang C, Jiang J, Jia C. 2019 Bacterial community shift in response to a deep municipal tail wastewater treatment system. *Bioresource Technol.* 281, 195–201. (DOI: 10.1016/j.biortech.2019.02.099)

The modifications are on yellow background.

5、 Some language and format issues should be revised carefully.

We thank the reviewer. We have improved the English by native speaker for English editing.

At the end, we would like to thank you again for the positive comments and constructive suggestions concerning our manuscript. Best regard to you!

Reviewer 2:

1. Heading Summary should be changed to Abstract.
2. For the introduction, a critical comparison of the adsorbents that were used previously for the removal of MB should be discussed to highlight their drawbacks/disadvantages that of new AC needs to be synthesized.
3. All chemicals used should be listed in sub 3.1 including the brand, city and country of origin.
4. Methylene blue should be in small letter unless used at the beginning of a sentence.
5. Sub 3.1 line 31-35 should be included in sub 3.2 as it was part of the AC synthesis procedures.
6. Sub 3.2 line 50: how can you confirm the ACFB synthesized was pure?
7. Sub 4.1: the authors should expand the characterizations to TGA and elemental analysis to prove the purity of ACFB.
8. For each parameter optimized, the author should indicate which optimum condition was chosen for the subsequent analysis. Another way is to indicate the batch adsorption method conditions in figure captions.

9. Sub 4.2.2 line 13: how is it possible to attain 100% removal?
10. For Fig 5, X-axis should be label as mass or amounts of ACFB (mg).
11. Sub 4.2.4: at the mentioned pH, what is the properties of ACFB? In this case, the change in ACFB surface needs to be discussed and correlate with ACFB properties.
12. R2 stands for correlation coefficient or non-linear regression coefficient?
13. The application of ACFB batch adsorption method for the removal of MB should be applied to the real sample to study the matrix effect and efficiency of ACFB in term of recovery.

Reply to Reviewer 2:

We thank and totally agree the reviewer for the comments and constructive criticism concerning our manuscript. We have improved our manuscript taking into account the remarks. Below are the replies for each specific comment:

- 1、 Heading Summary should be changed to Abstract.

We thank reviewer very much. We have changed the headline of ‘Summary’ to ‘Abstract’. The modifications are on yellow background.

- 2、 For the introduction, a critical comparison of the adsorbents that were used previously for the removal of MB should be discussed to highlight their drawbacks/disadvantages that of new AC needs to be synthesized.

We thank reviewer very much for this valuable suggestion. We have tried our best to discuss a critical comparison of various adsorbents previously using to remove MB and describe their drawbacks in the introduction part. The modifications are on yellow background.

- 3、 All chemicals used should be listed in sub 3.1 including the brand, city and country of origin.

We thank reviewer very much. We have revised and listed all chemicals we used in new sub 2.1 including their brand, city and country of origin. The modifications are on yellow background.

- 4、 Methylene blue should be in small letter unless used at the beginning of a sentence.

We thank and agree with the reviewer. We have revised all this mistake in the manuscript.

5、 Sub 3.1 line 31-35 should be included in sub 3.2 as it was part of the AC synthesis procedures.

We thank reviewer very much and agree with the reviewer. We have revised the content in sub 3.1 sub 3.2. The modifications are on yellow background.

6、 Sub 3.2 line 50: how can you confirm the ACFB synthesized was pure?

7、 Sub 4.1: the authors should expand the characterizations to TGA and elemental analysis to prove the purity of ACFB.

We thank reviewer very much for these valuable suggestion. We have modified the synthesis part in order to account for the obtained ACFB is pure, at the same time, the corresponding explanation is also made in sub 4.1, including the TGA, EA and XPS results from our previous studies.

- Luo L, Chen T, Li Z, Zhang Z, Zhao W, Fan M. 2018 Heteroatom self-doped activated biocarbons from fir bark and their excellent performance for carbon dioxide adsorption. J. CO2 Util. 25, 89-98. (DOI: 10.1016/j.jcou.2018.03.014)
- Zhao W, Luo L, Wu X, Chen T, Li Z, Zhang Z, Rao J, Fan M. 2019 Facile and low-cost heteroatom-doped activated biocarbons derived from fir bark for electrochemical capacitors. Wood Sci. Technol. 53, 227-248. (DOI: 10.1007/s00226-018-1065-3)

The modifications are on yellow background.

8、 For each parameter optimized, the author should indicate which optimum condition was chosen for the subsequent analysis. Another way is to indicate the batch adsorption method conditions in figure captions.

We thank the reviewer very much. We have revised the figure caption by adding the batch adsorption method conditions. The modifications are on yellow background.

9、 Sub 4.2.2 line 13: how is it possible to attain 100% removal?

We thank reviewer very much. We have tried our best to revise the manuscript, ‘~100%’ means an approximate value, the actually experimental data is 99.78%. We have revised this value in the paper on yellow background.

10、 For Fig 5, X-axis should be label as mass or amounts of ACFB (mg).

We thank reviewer very much and agree with the reviewer. We have done the modification, labeling the X-axis of figure 5 as mass of ACFB. The modifications are

on yellow background.

11、 Sub 4.2.4: at the mentioned pH, what is the properties of ACFB? In this case, the change in ACFB surface needs to be discussed and correlate with ACFB properties.

We thank reviewer very much for this valuable suggestion. We have tried our best to revise and discuss the surface properties change of ACFB in the part of ‘3.2.4 pH effect’. The modifications are on yellow background.

12、 R² stands for correlation coefficient or non-linear regression coefficient?

We thank the reviewer. R² stands for correlation coefficient.

13、 The application of ACFB batch adsorption method for the removal of MB should be applied to the real sample to study the matrix effect and efficiency of ACFB in term of recovery.

We thank reviewer very much and agree with the reviewer. We have tried our best to revise, supplement the data and add a new part of ‘4.4 Regeneration of adsorbent’ which we hope meet with your approval. The modifications are on yellow background.

At the end, we would like to thank you again for the positive comments and constructive suggestions concerning our manuscript. Best regard to you.